# Thermal Diffusivity of Concrete Samples Assessment Using a Solar Simulator

**DOI:** 10.3390/ma16031268

**Published:** 2023-02-02

**Authors:** Marcin Bilski, Przemysław Górnaś, Andrzej Pożarycki, Przemysław Skrzypczak, Mieczysław Słowik, Marta Mielczarek, Agnieszka Wróblewska, Łukasz Semkło

**Affiliations:** 1Institute of Civil Engineering, Faculty of Civil and Transport Engineering, Poznan University of Technology, 60-965 Poznan, Poland; 2Institute of Electrical Engineering and Electronics, Faculty of Automatic Control, Robotics and Electrical Engineering, Poznan University of Technology, 60-965 Poznan, Poland; 3Institute of Thermal Energy, Faculty of Environmental Engineering and Energy, Poznan University of Technology, 60-965 Poznan, Poland

**Keywords:** foam concrete, cement concrete, inverse problem, back-calculation, thermal properties, solar simulator, thermal diffusivity

## Abstract

The thermal properties of pavement layers made of concrete with varying bulk densities are a particularly interesting topic in the context of development road technologies. If a hybrid layer system is used as a starting point, with thin asphalt layers (from 1 cm to 4 cm) laid on top of a foam concrete layer, thermal properties begin to play a crucial role. The main research problem was to create a test method enabling the assessment of the influence of solar heating on the thermal parameters of the building material, especially cement concrete. For this reason, this paper is concerned specifically with the assessment of a new methodology for testing and calculating the value of the thermal diffusivity coefficient of samples made of concrete varying bulk densities. In this case, using the proprietary concept the authors built a solar simulator using a multi-source lighting system. The analysis of the results of laboratory tests and numerical analyses allowed the authors to observe that there is a strong correlation between the bulk density of samples heated and the thermal diffusivity parameter, which appears in the unidirectional heat transfer equation. The strength of this relationship has been expressed with the coefficient of determination and amounts to 99%. The calculated values of the coefficient of thermal diffusivity for samples made of foam concrete range from $0.16\times 10^{-6}$$\frac{m^{2}}{s}$ to $0.52\times 10^{-6}$$\frac{m^{2}}{s}$ and are lower (from 2.5 to 8 times) than the value determined for samples made of typical cement concrete.

## 1. Introduction

In concrete pavement design, the calculated total mechanical and thermal stresses determinate the dimensions of the slabs [1,2], which translates into construction and maintenance costs of these slabs [3,4]. Ongoing works on improving the technology of rigid pavements, carried out at least since the development of the first cement concrete pavement in 1892 (Ballefontaine, the US [5]), have led to the development of the standards currently in use. These standards make it possible to increase the dimensions of the slabs (driving down construction and maintenance costs, while providing better driving comfort due to a lower number of expansion joints) and to reduce their thickness (diminishing construction costs) by using different types of reinforcement and/or additives to the concrete mixes to improve their strength parameters [6,7]. Owing to, i.a., the results obtained in [8,9,10], the developed technology may provide an attractive alternative for slab reinforcement. The technology itself does not directly offer the effect of increasing the strength of the concrete layer (it may even decrease it), but makes it possible to reduce tensile stresses. Thermally induced stresses in the layer can be decreased by modifying the thermal properties of the concrete. This can prove a very effective solution, especially for local road pavements, where the dimensions of the slabs may be determined not by traffic loads, but rather by thermal stresses in the concrete layers. The required thicknesses or lengths of concrete pavement slabs often do not provide an economic justification for using such a construction solution. Among other things, this is why a large proportion of local roads are made in asphalt technology. In the case of local roads, the use of foam concrete as a base course is advantageous due to the properties reducing the load in the case of soils with low bearing capacity [11]. In the study cited above [8], adding glass to the concrete mix resulted in a significantly reduced temperature difference between the top and bottom surfaces of the slab, which directly translates into a reduction of thermal stresses arising in the pavement layer. Further research into the influence of glass additives or searching for other additives with the goal of improving thermal properties of pavement cement concretes requires a development of new testing methods. The authors [12] notice that what has a significant influence on concrete thermal properties such as thermal conductivity, coefficient of thermal expansion and specific heat is the mix design. Additionally, the type and content of a given aggregate significantly alters the thermal properties of the concrete it forms part of. Furthermore, the moisture condition of the concrete increases its specific heat and thermal conductivity. In the case of foam concrete, the air volume is related to the mixture composition and has an impact on thermal properties. The authors of the papers [13,14,15] presented the complex investigation of the thermo-mechanical behavior of cement concrete. For this purpose, the authors used traditional heating elements and numerical methods to describe the thermal parameters. When developing a new method, first and foremost, it seems vital to replace traditional heating elements [16] with a source that simulates solar radiation [17], which, among other things, will make it possible to analyse the effect the additive has on the value of the radiant energy absorption coefficient. It is an open research topic to develop test stands for heating concrete samples by means of a method similar to solar irradiation, coupled with a calculation method which can be used to calculate the values of the parameters determining the thermal properties of pavement cement concretes based on the obtained test results.

## 2. The Purpose and Scope of the Paper

The aim of this paper was the assessment a test method for calculating the value of the thermal diffusivity parameter of samples heated in a solar simulator. The simulator was the of the authors’ own design, built for the purpose of conducting studies in laboratory conditions. The computational method developed in the study allows for the parametric determination of the thermal properties of cement concrete. The method is based on the idea of the so-called inverse problem [18,19], which is solved by calculating the values of the physical parameters of the modelled object. The inverse problem as a mathematical scientific issue is the theoretical basis of the calculations used in paper. For the purposes of this article, this calculation is called the back-calculation. In order to achieve the stated objective, the following scope of work was adopted:preparation of construction of a test bench for simulating solar radiation in laboratory conditions,laboratory testing in solar simulator of cubic samples made of cement concrete varying in density, which show substantial differences with respect to thermal properties,development of a computer implementation method of back-calculation procedure for the determination of thermal diffusivity coefficient values,performing regression analysis on sets containing results of calculated values of parameters defining thermal properties of concrete samples varying bulk density,discussion and conclusions.

## 3. Materials and Methods

### 3.1. Testing Materials

Tests on the thermal properties were conducted on cubic 15 × 15 × 15 cm samples made of regular cement concrete and foam concrete. The samples differing in terms of their bulk density were made in accordance with standard [20] and with the principles described in works [11,21,22]. By use of the recipes for concrete mixes used in the laboratory, twenty samples were prepared—four samples for each bulk density value. The authors obtained the following mean bulk density values (with uncertainty) of bulk density ρ of concrete samples:–ρ = 2319±3$\frac{\mathrm{kg}}{m^{3}}$—cement concrete C45/55 ($f_{ck,cube}$ = 56.8±0.8 MPa),–ρ = 1173±2$\frac{\mathrm{kg}}{m^{3}}$—foam concrete LDCC${}_{1200}$ ($f_{ck,cube}$ = 5.9±0.2 MPa),–ρ = 773±5$\frac{\mathrm{kg}}{m^{3}}$—foam concrete LDCC${}_{800}$ ($f_{ck,cube}$ = 2.5±0.2 MPa),–ρ = 636±11$\frac{\mathrm{kg}}{m^{3}}$—foam concrete LDCC${}_{650}$ ($f_{ck,cube}$ = 2.1±0.1 MPa),–ρ = 434±9$\frac{\mathrm{kg}}{m^{3}}$—foam concrete LDCC${}_{400}$ ($f_{ck,cube}$ = 1.4±0.1 MPa).

### 3.2. Test Bench

The test bench, part of which is the solar radiation simulator, was made taking into account the standard spectral parameters of solar radiation. In accordance with the requirements presented in Table 1 concluded in the document [23], the global reference solar irradiance distribution for individual wavelength sub-ranges has been determined. Classification of simulators is presented in Table 2 “Definition of solar simulator classifications” concluded in the document [23]. There are three classes A, B, C, for which the maximum possible percentage errors are given (relative errors expressed as a percentage) in relation to the specified spectrum of radiation presented in Table 1 concluded in the document [24]. Using a multi-source system, it is possible to adjust the control of individual radiation sources. This approach makes it possible to obtain the highest class of the spectral projection. The created test bench for heating cubic samples meets the conditions for simulators in the highest accuracy class A and consists of five components:–a heat emission source simulating solar radiation—the upper part of the test bench,–a thermal isolation part—the bottom part of the test bench,–a set of five temperature measurement transducers,–a data acquisition module,–a computer module for controlling, processing and archiving test results.

The heat emission source (solar simulator) was built in the form of a metal dome shown in Figure 1 with five GU10 halogen bulbs of 50 W each with IR filters and light-emitting diodes (LEDs) with different wavelengths of emitted radiation shown in Figure 2a.

In order to match the spectral distribution of the simulator to the requirements described in standard [25], the diode set was equipped with a constructed thyristor current control system, shown in Figure 2b. The radiation distributions obtained in the simulator set against the solar radiation distribution are shown in Figure 3.

Due to the fact that the testing methodology assumes inducing a unidirectional heat flow from the top surface of the sample to its bottom it was necessary to limit the heat transfer between the sample and the environment. These conditions, as determined above, have been provided by equipping the test bench shown in Figure 4a with thermally isolated chamber (made with mineral wool and aluminum foil) and the steel plate designed to reflect light rays (only the upper part of the sample is exposed to direct radiation) shown in Figure 4b.

A set of five thermoelectric transducers (type K thermocouples) was connected to a temperature acquisition unit presented in Figure 4a and Figure 5a and used to record the temperature changes at the height of the heated sample. The design of the acquisition unit has been based on a microcontroller embedded in the MEGA2560 R3 integrated circuit with ATMega2560 16 MHz microcontroller and CH340 programmer. A set of five thermoelectric transducers MAX6675 with SPI interface is used to process the signals. In the Figure 5b the direction of heat flow during the sample heating is shown with a yellow arrow and the specific transducers are marked from T1 to T5 with red color. The transducers are arranged vertically, i.e., thermocouple no. 1 (mark T1) is on the top of the sample (closest to the heated surface—1.5 cm) and thermocouple no. 5 (mark T5) is on the bottom of the sample (farthest from the heated surface—13.5 cm). The transducers were placed in holes drilled in the sample and guarantee the location of transducer’s measuring element in its axis.

Along with the design of the data acquisition module, the authors built a circuit that was used to capture the measurement values from all the transducers and send them to a Universal Serial Bus (USB). The computer code listing of the main loop in the acquisition algorithm for the ATMega2560 microcontroller instructions used in the test bench presented by Listing 1.

**Listing 1.** The main loop in the acquisition algorithm.

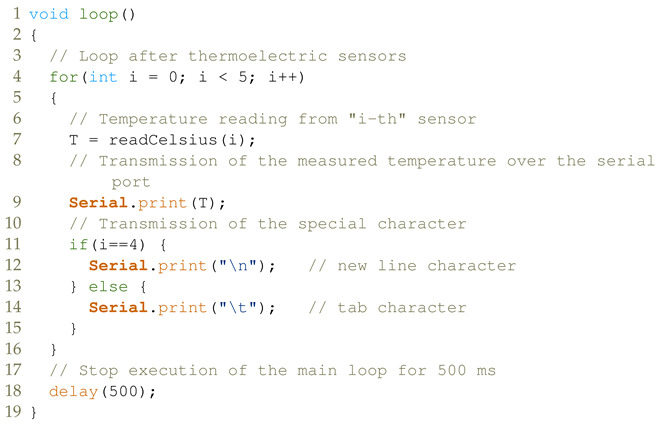



For the purpose of recording data generated in the solar simulator, the authors prepared an application written in C++ for the Ubuntu Linux distribution (Ubuntu version 20.04) to capture the values measured from the USB port and save them to the computer’s hard drive. Prior to testing, the thermoelectric transducers (thermocouples) were calibrated using a calibrated thermometer to obtain the required measurement accuracy of ±0.25 °C.

### 3.3. Verification of Unidirectional Heat Flow through the Sample

To check against the assumption of a unidirectional heat flow through the sample a cubic sample of cement concrete was placed in the test bench and subjected to heating. During the process of heating, at 10-minute intervals, a temperature reading was taken on one of the side walls of the sample by use of a thermal imaging camera. Figure 6 shows an example of the results of this measurement, the analysis of which confirmed that the heat flow through the sample could be considered unidirectional and therefore in accordance with the assumptions.

### 3.4. Laboratory Experiment Using a Solar Simulator

In order to determine the change in temperature value at different altitudes marked in Figure 5b for the samples presented in point Section 3.1, they were heated in the test bench for a period of two hours. The authors noticed that after more than an hour and a half upon the start of heating, there is a state in which the defined steady increase in the temperature value occurs in individual thermocouples. Thus, the period of 7000 s is fully sufficient to perform the analysis of the thermal properties of the tested material and obtain unambiguous values of parameters using back-calculations. An example of the results obtained during heat a sample in the solar simulator is shown in the Figure 7.

The temperature values obtained in this test were used in the backcalculattion computational method described in the next subsection for the determination of the thermal diffusivity coefficient.

### 3.5. Back-Calculation Computational Method

Back-calculations are formulated in form of iteration. Subsequent iterations are used to find values for the model parameters, so that the difference between the measurement results and the model responses tends to a minimum. Algorithm consists of (1) a computational model, (2) an optimisation algorithm and (3) an error function. The selection of these three elements has a direct impact on the calculation results obtained.

In this paper a computational model of the inverse problem was constructed based on the numerical solution of the Fourier differential equation. This equation forms the basis of the mathematical description of heat flow in the modelled sample in the test bench, and the form adopted for consideration is expressed in Equation (Equation 1).
(1)$$\frac{\partial T(t)}{\partial t}=a\cdot \nabla^{2}\cdot T\left(t\right)$$
where: T(t)—temperature function in time [K], *t*—time [s], *a*—thermal diffusivity coefficient expressed in Equation (Equation 2) [$\frac{m^{2}}{s}$], $\nabla^{2}$—Laplace operator.
(2)$$a=\frac{\lambda }{c_{p}\cdot \rho }$$
where: λ—thermal conductivity coefficient [$\frac{W}{m\cdot K}$], $c_{p}$—specific heat [$\frac{J}{\mathrm{kg}\cdot K}$], ρ—density [$\frac{\mathrm{kg}}{m^{3}}$].

The Equation (Equation 1) was solved by use of a finite difference method [26,27]. The initial and boundary conditions were adopted based on conditions occurring during testing of the sample in the test bench. Based on confirmed by a laboratory experiment (presented in Section 3.3) the one-way flow of thermal energy, one-dimensional numerical model was used. With reference to Figure 8, the assumed initial condition for all nodes from *i* to *n* with a constant distance between them (dx = 1 mm) is expressed in Equation (Equation 3).
(3)$$T_{i}\left(x_{i},t\right)=T_{i+1}\left(x_{i+1},t\right)=\ldots =T_{n-1}\left(x_{n-1},t\right)=T_{n}\left(x_{n},t\right)=T_{k}$$
where: *t*—time variable [s], $T_{i}\left(x_{i},t=0\right)\left[K\right]$—initial temperature of the *i*-th node along the *x* axis, $T_{k}\left[K\right]$—temperature of the concrete sample after 2 h of conditioning at environment temperature (from 293 K to 296 K i.e., from 20 °C to 23 °C) immediately before starting the heating of the sample.

From the set of all nodes, five of them i.e., T1(*t*), T2(*t*), T3(*t*), T4(*t*) and T5(*t*) were assigned a geometric location that corresponded to the location of thermocouples in the concrete sample as shown in Figure 8. It is assumed that the model of thermal energy exchange between the sample and the environment is realized through the extreme nodes. The temperature loading values for T1(*t*) and T5(*t*) were assumed to be equal to those measured when the sample was heated. Values of the other ones T2(*t*), T3(*t*) and T4(*t*) were back-calculated.

Upon the analysis conducted by the authors, it was found that the concept of the inverse problem can be used to unequivocally determine the values of the *a* parameter in the model, i.e., the values of the thermal diffusivity coefficient. In the paper as a starting point the geometry of horizontal building partitions made of cement concrete placed on a layered half-space was taken. The boundary conditions of the laboratory stand related to the phenomena of emissivity, penetration and reflection of heat energy waves were determined in a method similar to real conditions, namely:a unidirectional flow of thermal energy is ensured in the direction from the heated surface of the sample to its lower surface,the filling of the components of the solar radiation spectrum is realized in a simulator with a multi-diode and incandescent combination of the highest class of spectral projection,the exchange of heat energy between the sample and the environment takes place in a space with the volume of the dome of the test bench, which imitates the actual vicinity of the horizontal building partition with sunlight.

A natural consequence of the proposed approach is a one-dimensional numerical model. For the being heated sample in the test bench, Figure 8 shows the adopted schematic. In this figure markers T1 => T1(*t*), *…*, T5 => T5(*t*) mean individual transducers (type K thermocouples).

The calculations are reduced to calculating the value of the *a* parameter with the smallest difference between the temperature values T2(t), T3(t) and T4(t), measured as a function of time, and the corresponding temperature values calculated by means of the numerical model.

In order to determine the value of the *a* parameter, the Nelder Mead optimisation algorithm [28] was used in the inverse problem. The assessment of the match of the model results and those recorded during testing was performed by use of the root mean square error (RMSE), described by Equation (Equation 4).
(4)$$RMSE=\sqrt{\frac{1}{n}\cdot \overset{n}{\sum_{i=1}}{\left(T_{i}-\hat{T_{i}}\right)}^{2}}\cdot 100\%$$
where: *n*—the number of measured and calculated temperature values [−], $T_{i}$—*i*-value of the measured temperature from test, $\hat{T_{i}}$—*i*-value of the calculated temperature with the model.

## 4. Results of Heating Test in Solar Simulator

The Figure 9 shows diagrams illustrating the test results (mean values) obtained for all heating samples using a solar simulator with duration of 7000 s. Based on paper [29], it can also be presupposed that the thermal properties of the samples prepared for testing in the respective series can differ in a statistically significant way. The analysis of the results presented in Figure 9, allows for the observation that with the decrease in the value of the sample’s bulk density, the heating of the upper surface of the sample is faster and greater (T1(*t*) thermocouple—away from the heated surface by 1.5 cm). A slower and smaller temperature rise in the lower part of the sample along with the decreasing of its bulk density can also be noticed (T4(*t*) and T5(*t*) thermocouples – away from the heated surface by 10.5 cm and 13.5 cm). The diagrams show that in the case of typical cement concrete, there is a fairly even heat flow through the sample (uniform heating of the entire sample volume), and in the case of foam concrete, the upper part is quickly heated and the heat flows gradually towards the lower part.

The *a* thermal diffusivity coefficient [$\frac{m^{2}}{s}$] is used to show the relationship that exists between the temperatures inside the sample and its surface. The value of this coefficient should be associated with the susceptibility of the material to thermal diffusivity involving the temperature inside the sample and the temperature during the process of heating or cooling of its surface. The greater the value of the *a* coefficient, the faster the change in the sample’s internal temperature under the influence of the temperature change in the solar simulator. The calculation results obtained using the back-calculation procedure are summarized in Table 1. The analysis of the table shows that the values of the thermal diffusivity coefficient *a* decrease as the bulk density of concrete samples is reduced. Thermal diffusivity defines the rate of transfer of heat of a material from the hot end to the cold end. This means that the smaller the bulk density of a concrete sample, the slower the heat transfer inside it. This confirms that the temperature values presented in the Figure 9 and the thermal diffusivity coefficient values obtained through the use of the back-calculation. The results confirm the insulating properties of foam concrete due to the high content of air bubbles.

## 5. Discussion

The thermal diffusivity coefficient forms an integral part of the unidirectional heat transfer equation. In the paper it was used to analyze the susceptibility of a cubic sample made of concrete to thermal diffusivity between the interior of the sample through its top surface and laboratory solar simulator—which simulates heating due to solar radiation.

The results of the conducted tests confirm that the developed method can be used to determine the value of thermal diffusivity coefficient *a* and confirm its dependence on the value of concrete bulk density. The obtained values of *a* parameter do not differ significantly from the values presented in the publication [30]. For example, the value of the thermal diffusivity coefficient determined by the authors of this present paper equals for foam concrete 636 $\frac{\mathrm{kg}}{m^{3}}$ bulk density a=$0.26\times 10^{-6}$$\frac{m^{2}}{s}$ and determined in [30] for foam concrete 600 $\frac{\mathrm{kg}}{m^{3}}$ bulk density a=$0.35\times 10^{-6}$$\frac{m^{2}}{s}$. The composition and content of air bubbles have a large impact on the obtained values of *a* parameter, and therefore it was necessary to make identical samples to the ones used by the authors of the publication [30] in order to determine the specific difference. The values obtained at this stage of the work are of the same order. The analysis of the test results confirms that the methodology presented in this paper makes it possible to determine the relationship between the bulk density of samples heated in the simulator and the *a* parameter of the unidirectional heat flow equation. In terms of statistics, the Figure 10 illustrates the regression analysis in which the strength of the sought correlation relationship between these quantities, expressed by the value of the coefficient of determination, amounts to 99%. These equation can be used to calculate the value of *a* parameter depending on the bulk density of the foam concrete.

With regard to the construction of the developed test bench, it should be emphasized that the solar radiation simulators built and used so far were built on the basis of Xenon sources [31,32] or sulfur lamps [33]. In the created solar simulator, a multi-diode combination (with different spectral distributions) was used together with incandescent sources to fill the spectrum in the infrared range and a set of infrared filters. Using this solutions it was possible to correct the control of individual radiation sources, which allowed for the highest class of the spectral projection. Created test bench also has useful practical features. In the case of changing the parameters of individual light sources during exploitation, it is possible to replace a single source. After rescaling the research can be continued. The disadvantage of using, inter alia, light-emitting diodes is the limitation of the power of the sources, which was associated with the limitation of the surface area on which the generated level of irradiation (1000 W/m${}^{2}$) is maintained and complies with the conditions set out in [23] (p. 8, pt. 4).

It is worth noting the thermal properties of roadway pavement made in hybrid technology—a foam concrete subbase with thin (from 2 cm to 4 cm) asphalt layer. The results of the research allow to assume that the foam concrete layer will behave as a heat receiver. The asphalt layers, which are sensitive to temperature changes, gain a layer in this hybrid system that can act as a heat sink. Even though this thesis requires further verification in laboratory and in-site studies, the values obtained for the *a* diffusivity coefficient justify this line of research.

## 6. Conclusions

This paper outlined a research study in which the authors focused on developing a uniform method for determining values of the thermal diffusivity coefficient in materials used in roadway pavement layers. The results of the performed tests, combined with a computer algorithm providing a solution to the inverse problem, constitute the basis for the determining method described in this paper and enable the determination of this coefficient. Laboratory tests were carried out using a solar simulator of the authors’ own design. The calculated values of the thermal diffusivity coefficient for samples made of foam concrete range from $0.16\times 10^{-6}$$\frac{m^{2}}{s}$ to $0.52\times 10^{-6}$$\frac{m^{2}}{s}$. For a sample with a bulk density of 434 $\frac{\mathrm{kg}}{m^{3}}$ the value of the *a* parameter is approximately eight times lower than the value of that parameter for a sample made of cement concrete C45/55 with a bulk density of 2319 $\frac{\mathrm{kg}}{m^{3}}$. A smaller value of this coefficient indicates that a layer featuring this coefficient will warm up at a slower rate. The regression analysis proved that the value of the diffusivity coefficient strongly depends on the bulk density of the sample. The value of the coefficient of determination $R^{2}=99\%$. This leads to the hypothesis that the temperature distribution in roadway pavement layers can be controlled by selecting a system of layers made of concrete of different bulk densities in conjunction for example with compressive strength. It is noteworthy that the results of this research can be used to design hybrid structural systems based on a combination of thin asphalt layer and foam cement concrete subbase. However, the development of specific guidelines in this area requires further research work. The advantages of the developed test method include the use of a multi-source lighting system and obtaining the highest class spectral projection. From the point of view of numerical modeling it is enabled the use of relatively simple mathematical models. The construction of the solar simulator enables the replacement of a single light source, and after the rescaling operation, the research can be continued. The disadvantage of using light-emitting diodes is the limitation of the power of the sources, which is associated with the limitation of the surface area of the sample in the simulator. The presented test method will be used in the future by the authors in research related to the searching for additives changing the thermal properties of cement concretes.

## Figures and Tables

**Figure 1 materials-16-01268-f001:**
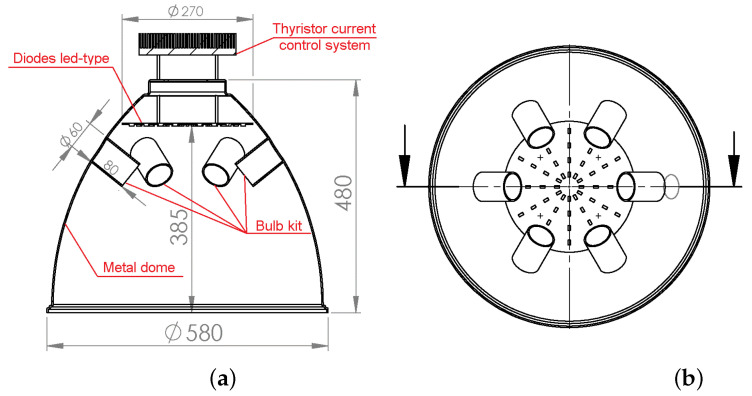
Solar simulator (the upper part of the test bench): (**a**) cross-section, (**b**) bottom view.

**Figure 2 materials-16-01268-f002:**
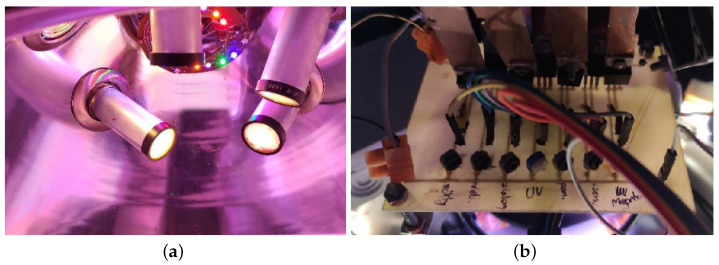
Solar simulator: (**a**) the system of heat emission sources, (**b**) thyristor.

**Figure 3 materials-16-01268-f003:**
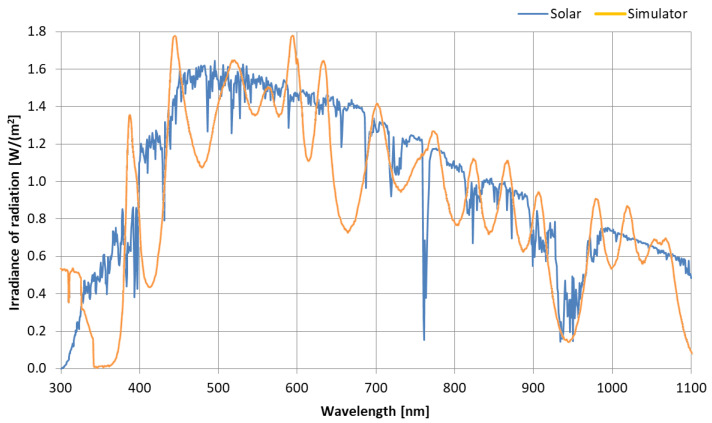
The radiation distribution obtained in the test bench set against the solar radiation distribution.

**Figure 4 materials-16-01268-f004:**
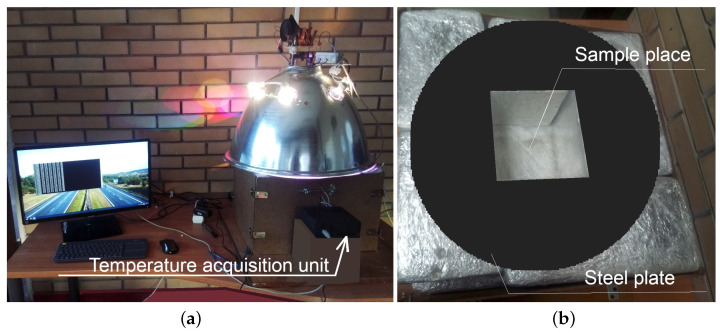
The test bench: (**a**) overall view, (**b**) part with the thermal insulation and the steel plate.

**Figure 5 materials-16-01268-f005:**
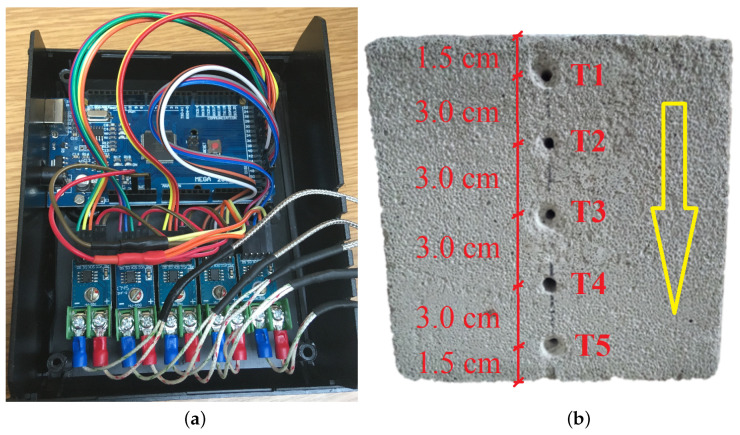
(**a**) The subassemblies inside the temperature acquisition unit (**b**) Schematic of the arrangement of the temperature measuring transducers in holes drilled in the sample.

**Figure 6 materials-16-01268-f006:**
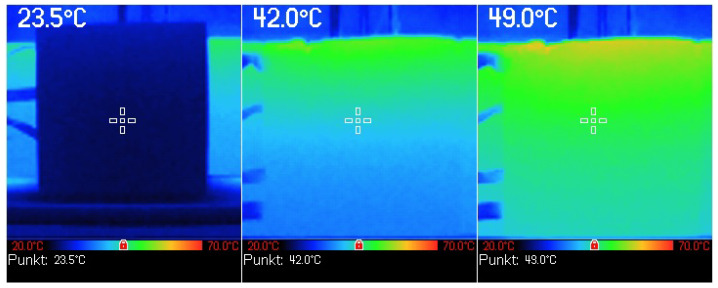
Illustration of the heat flow direction, based on the study of one of the side walls of a sample heated in the test bench using a thermal imaging camera.

**Figure 7 materials-16-01268-f007:**
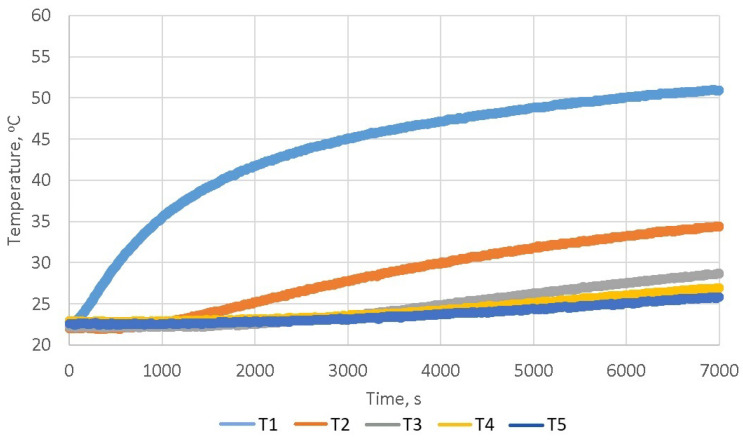
The dependence of the temperature value on the time in individual thermocouples during heating a concrete sample.

**Figure 8 materials-16-01268-f008:**
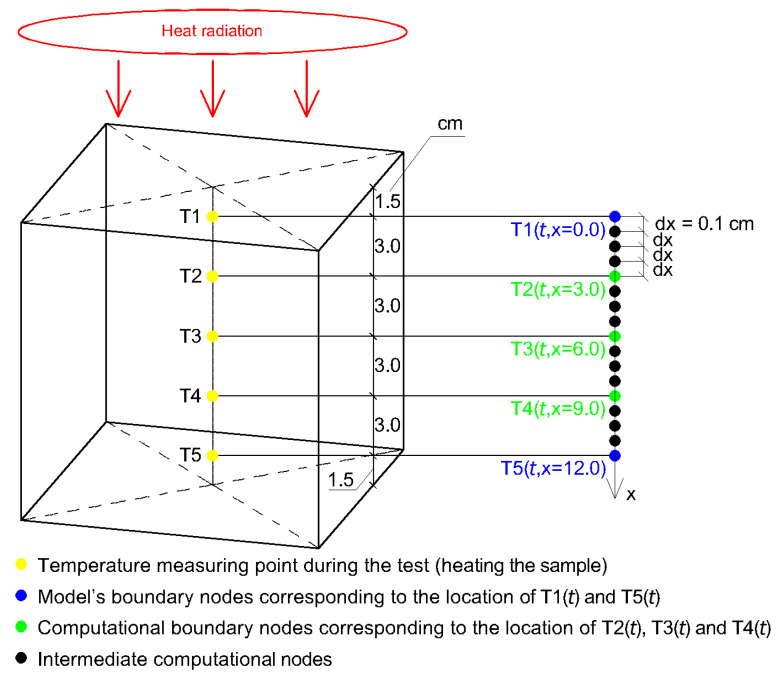
Schematic of the numerical model.

**Figure 9 materials-16-01268-f009:**
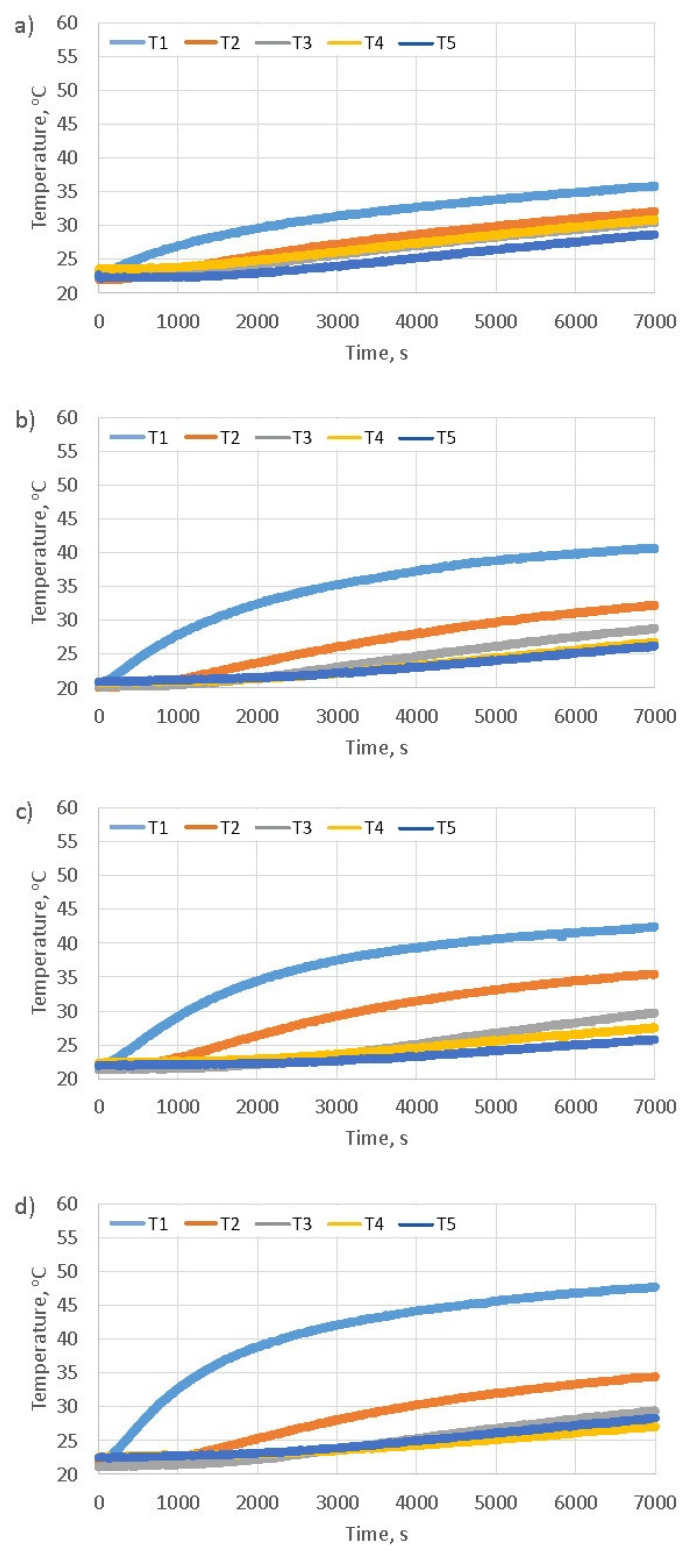
Diagrams of temperature dependence for individual thermocouples T1(*t*), T2(*t*), T3(*t*), T4(*t*), T5(*t*) on the heating time of the concrete samples varying bulk density: (**a**) 2319 $\frac{\mathrm{kg}}{m^{3}}$, (**b**) 1173 $\frac{\mathrm{kg}}{m^{3}}$, (**c**) 773 $\frac{\mathrm{kg}}{m^{3}}$, (**d**) 636 $\frac{\mathrm{kg}}{m^{3}}$, (**e**) 434 $\frac{\mathrm{kg}}{m^{3}}$.

**Figure 10 materials-16-01268-f010:**
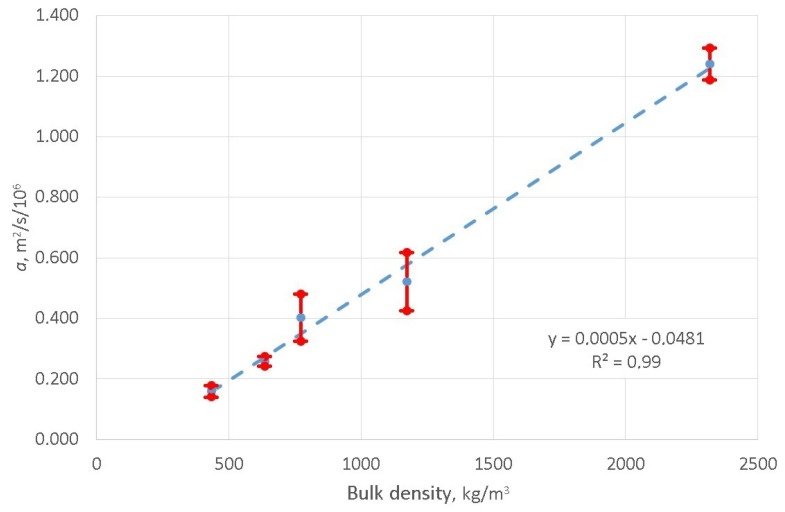
Graph showing the relationship between the bulk density of concrete samples and the *a* parameter values.

**Table 1 materials-16-01268-t001:** Summary of values for the thermal diffusivity coefficient calculated using the back-calculation procedure.

Sample No.	Bulk Density	RMSE	*a*
	[$\frac{\mathrm{kg}}{m^{3}}$]	[%]	[$\frac{m^{2}}{s}$]
1.1	2318	2.98	1.273 $\times 10^{-6}$
1.2	2322	3.80	$1.236\times 10^{-6}$
1.3	2317	4.44	$1.218\times 10^{-6}$
1.4	2321	1.84	$1.233\times 10^{-6}$
*mean value/uncertainty*	2319 ± 3		(1.240 ± 0.053)$\times 10^{-6}$
2.1	1175	1.47	$0.477\times 10^{-6}$
2.2	1172	1.09	$0.524\times 10^{-6}$
2.3	1174	1.30	$0.506\times 10^{-6}$
2.4	1172	1.16	$0.578\times 10^{-6}$
*mean value/uncertainty*	1173 ± 2		(0.520 ± 0.096)$\times 10^{-6}$
3.1	772	2.00	$0.448\times 10^{-6}$
3.2	776	1.22	$0.410\times 10^{-6}$
3.3	777	1.88	$0.370\times 10^{-6}$
3.4	767	2.61	$0.381\times 10^{-6}$
*mean value/uncertainty*	773 ± 5		(0.402 ± 0.078)$\times 10^{-6}$
4.1	636	2.42	$0.259\times 10^{-6}$
4.2	647	2.37	$0.250\times 10^{-6}$
4.3	640	2.71	$0.258\times 10^{-6}$
4.4	621	2.48	$0.268\times 10^{-6}$
*mean value/uncertainty*	636 ± 11		(0.259 ± 0.017)$\times 10^{-6}$
5.1	425	3.13	$0.165\times 10^{-6}$
5.2	442	3.50	$0.167\times 10^{-6}$
5.3	443	1.82	$0.145\times 10^{-6}$
5.4	428	4.00	$0.160\times 10^{-6}$
*mean value/uncertainty*	434 ± 9		(0.160 ± 0.019)$\cdot 10^{-6}$

## Data Availability

Not applicable.

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
