# Peer review of "Thermal Diffusivity of Concrete Samples Assessment Using a Solar Simulator"

_materials, 2023, doi:10.3390/ma16031268_

Round 1
Reviewer 1 Report
This contribution is focused on the determination of the thermal diffusivity coefficients of concrete pavements. The desired material property is identified from the proposed algorithmic framework minimizing the difference between the experimentally measured and computationally obtained temperature curves. The text is well-structured and after answering my comments it is mature enough for publication.
General comments:
i) chapter 3.5, the introduction of boundary and initial conditions in the sense of mathematical formulas and parameter values is missing in this chapter. The statement about setting the boundary conditions the same as in the experimental setup is insufficient. What about initial conditions? The scheme displaying the boundary and initial conditions entering the numerical analysis needs to appear in this chapter. Moreover, if I understand correctly Figure 10, the radiation boundary condition is introduced in your calculations, but the value of emissivity of the surface is not mentioned. Your inverse analysis is probably performed for a fixed value of this boundary coefficient and this is again not commented on. Since this value is uncertain, It should be introduced in the inverse analysis as another parameter that needs to be identified. I am also missing another type of boundary condition such as heat transfer bringing another parameter - heat transfer coefficient - to play.
Minor comments:
i) l. 43, remove right bracket
ii) l. 53, correct word coefficient
iii) l.147, correct word backcalculation. Consider the proper writing such as back-calculation.
iv) l. 181 and multiple occurrences in the figures, the variable time $t$ should be written in italic font. Consider this correction also for the temperature values T1 - T5.
v) I appreciate the different works you have done, but Figures 6 and 7 are not necessary to show in your manuscript. These are simple procedures that do not need to be mentioned.
Reviewer 2 Report
Dear authors of the manuscript entitled “ASSESSMENT OF THE THERMAL DIFFUSIVITY OF CONCRETE SAMPLES VARYING BULK DENSITIES USING A SOLAR SIMULATOR”,
I have carefully reviewed your manuscript. The topic seems fit with the scope of the journal. However, you are required to address the following issues before considering the manuscript for publication:
1- Please re-write the title of research according to journal style.
2- There are some typing and grammar mistakes. Therefore, I suggest reviewing the entire manuscript by a professional editing service.
3- In the abstract, please describe briefly the methodology conducted in the research. As, the abstract section should be one paragraph up to (250) words, summarizes the major aspects of the entire paper in a prescribed sequence that includes: 1) the research problem; 2) the overall purpose of the research; 3) the methodology and/or data used; 4) major findings or the important outcomes of the research. Therefore, please revise accordingly.
4- More recent literature should be included in the introduction to add power to the research. Therefore, please consider revision.
5- References are not all in the same format. Therefore, please use proper tool for citation (e.g. Endnote or Mendeley) and follow the journal style.
6- Please download the citation from original source (e.g science direct, springer, …etc.). not simply use google scholar. Moreover, it is strongly encouraged to include the DOI of references.
7- Finally, it has been noticed there is not any connection between your research and articles already published in this journal. Please consider this matter.
Therefore, the reviewer believes that the manuscript is not suitable for publication in its present form due to the above reasons
Reviewer 3 Report
In this paper, the authors have tried to develop a strong correlation between the bulk density of samples heated and the thermal diffusivity parameter of the concrete with varying densities. I have some comments regarding this manuscript (MS).
1. More related citations should be included in the Introduction.
2. How this work differs from the reported work?
3. Novelty of the work should be highlighted.
4. What are the advantages and disadvantages of this method?
5. The results obtained in this work should be compared with the reported data.
6. Discussion is concise, it should be provided in detail.
7. The conclusion should be reconstructed. E.g. methodology adopted main outcomes, advantages, and future impact.
Reviewer 4 Report
The paper could address the limitations of the study, such as any potential sources of bias or error that may have affected the results. It could also discuss the generalizability of the findings and how they might apply to other types of pavement materials or real-world conditions.
It is not clear from the information provided how the authors have concluded that their test bench meets the conditions for simulators in the highest accuracy class, or what criteria they have used to make this determination. In order to fully understand and evaluate this claim, it would be helpful to have more information about the specific standards or guidelines used to define the accuracy class of simulators and how the authors' test bench meets these requirements.
I would like to recommend to the authors that they consider including the computer code of the main loop in the acquisition algorithm, which is currently presented between lines 123-124, as an appendix at the end of the article. It would be helpful to refer to this code in the main text, rather than having it pasted as an image. Additionally, I would suggest that the quality of this image be improved or that the code be presented in text format for better readability. The same issue for Figure 7.
Also, I suggest to the authors that they consider improving the quality of figures 9, 11, and 12. These figures may be difficult to interpret due to their low quality. In addition, it would be helpful if these figures included legends to provide context and clarify the meaning of the data being presented.
Round 2
Reviewer 2 Report
Dear authors of the manuscript entitled “Thermal Diffusivity of Concrete Samples Assessment Using a Solar Simulator”,
I have carefully reviewed your revised manuscript and have concluded the following remarks:
1- The manuscript is generally well-revised with a comprehensive introduction, adequate methodology, and an intense discussion of the results including tables, figures, diagrams, …etc.
2- Author responses to reviewers’ comments are well addressed.
Author Response
We are very appreciate for your revision. Thank you very much. Best Regards.
Reviewer 3 Report
In this paper, the authors have tried to develop a strong correlation between the bulk density of samples heated and the thermal diffusivity parameter of the concrete with varying densities. I have some comments regarding this manuscript (MS).
1. More related citations should be included in the Introduction.
2. How this work differs from the reported work?
3. Novelty of the work should be highlighted.
4. What are the advantages and disadvantages of this method?
5. The results obtained in this work should be compared with the reported data.
6. Discussion is concise, it should be provided in detail.
7. The conclusion should be reconstructed. E.g. methodology adopted main outcomes, advantages, and future impact.
Please provide a response to the above comments.
Author Response
Dear Reviewer. For the rest of the received reviews, we have positive feedback after making the corrections. Only in your review, we have not received comments to the corrected version. We have achieved exactly the same ones as in the first review, hence we are sending clarifications again. 1. More related citations should be included in the Introduction. We have made corrections in the introduction. 2. How this work differs from the reported work? We have made corrections in the section 3.2. 3. Novelty of the work should be highlighted. We have made corrections in the discussion and conclusions. 4. What are the advantages and disadvantages of this method? We have made corrections in the discussion and conclusions. 5. The results obtained in this work should be compared with the reported data. We have made corrections in the introduction, section 3.2., discussion and conclusions. 6. Discussion is concise, it should be provided in detail. We have made corrections in the in discussion. 7. The conclusion should be reconstructed. E.g. methodology adopted main outcomes, advantages, and future impact. We have made corrections in the conclusion. Best Regards